# Transcriptional Profile Associated with Clinical Outcomes in Metastatic Hormone-Sensitive Prostate Cancer Treated with Androgen Deprivation and Docetaxel

**DOI:** 10.3390/cancers14194757

**Published:** 2022-09-29

**Authors:** Natalia Jiménez, Òscar Reig, Mercedes Marín-Aguilera, Caterina Aversa, Laura Ferrer-Mileo, Albert Font, Alejo Rodriguez-Vida, Miguel Ángel Climent, Sara Cros, Isabel Chirivella, Montserrat Domenech, Mariona Figols, Enrique González-Billalabeitia, Daniel Jiménez Peralta, Leonardo Rodríguez-Carunchio, Samuel García-Esteve, Marta Garcia de Herreros, Maria J. Ribal, Aleix Prat, Begoña Mellado

**Affiliations:** 1Translational Genomics and Targeted Therapeutics in Solid Tumors Lab, Institut d’Investigacions Biomèdiques August Pi i Sunyer (IDIBAPS), 08036 Barcelona, Spain; 2Fundació Clínic per a la Recerca Biomèdica, 08036 Barcelona, Spain; 3Medical Oncology Department, Hospital Clínic, 08036 Barcelona, Spain; 4Uro-Oncology Unit, Hospital Clínic, University of Barcelona, 08036 Barcelona, Spain; 5Department of Medicine, University of Barcelona, 08036 Barcelona, Spain; 6Medical Oncology Department, Institut Català d’Oncologia, Hospital Germans Trias i Pujol, 08916 Badalona, Spain; 7Medical Oncology Department, Institut Hospital del Mar d’Investigacions Mèdiques (IMIM), Hospital del Mar, 08003 Barcelona, Spain; 8Medical Oncology Service, Instituto Valenciano de Oncología (IVO), 46009 Valencia, Spain; 9Medical Oncology Department, Hospital General de Granollers, 08402 Granollers, Spain; 10Oncology Department, Hospital Clínico Universitario de Valencia, 46010 Valencia, Spain; 11Medical Oncology Department, Fundació Althaia Manresa, 08243 Manresa, Spain; 12Medical Oncology Department, Hospital Universitario 12 de Octubre, 28041 Madrid, Spain; 13Urology Department, Hospital General Universitario José M. Morales Meseguer, 30008 Murcia, Spain; 14Department of Pathology, Hospital Clínic, 08036 Barcelona, Spain

**Keywords:** metastatic prostate cancer, predictive biomarkers, hormonal therapy, chemotherapy, androgen receptor, estrogen receptor, tumor suppressor genes

## Abstract

**Simple Summary:**

The combination of androgen deprivation therapy (ADT) with docetaxel (DX) or/and with novel anti-androgen receptor therapies have become standards for the treatment of patients with metastatic hormone-sensitive prostate cancer (mHSPC). However, metastatic PC remains incurable, and biomarkers for individual treatment selection are needed. We propose here that molecular alterations associated with castration resistance may predict the clinical evolution of mHSPC patients. To test this hypothesis, we designed a custom expression panel of 184 genes and tested it in tumor biopsies from patients with mHSPC treated with ADT+DX. We found that AR and ESR signatures and *ESR2* gene expression correlate with a good prognosis. The lower expression of TSG (*PTEN*, *TP53* and *RB1*) signature, as well as high *ARV7* and low *RB1* gene expression, were associated with adverse clinical outcomes. The usefulness of transcriptomic analysis of such signatures as a strategy for personalized treatment selection should be further explored.

**Abstract:**

(1) Background: Androgen deprivation therapy (ADT) and docetaxel (DX) combination is a standard therapy for metastatic hormone-sensitive prostate cancer (mHSPC) patients. (2) Methods: We investigate if tumor transcriptomic analysis predicts mHSPC evolution in a multicenter retrospective biomarker study. A customized panel of 184 genes was tested in mRNA from tumor samples by the nCounter platform in 125 mHSPC patients treated with ADT+DX. Gene expression was correlated with castration-resistant prostate cancer-free survival (CRPC-FS) and overall survival (OS). (3) Results: High expression of androgen receptor (AR) signature was independently associated with longer CRPC-FS (hazard ratio (HR) 0.6, 95% confidence interval (CI) 0.3–0.9; *p =* 0.015), high expression of estrogen receptor (ESR) signature with longer CRPC-FS (HR 0.6, 95% CI 0.4–0.9; *p =* 0.019) and OS (HR 0.5, 95% CI 0.2–0.9, *p =* 0.024), and lower expression of tumor suppressor genes (TSG) (*RB1*, *PTEN* and *TP53*) with shorter OS (HR 2, 95% CI 1–3.8; *p =* 0.044). *ARV7* expression was independently associated with shorter CRPC-FS (HR 1.5, 95% CI 1.1–2.1, *p =* 0.008) and OS (HR 1.8, 95% CI 1.2–2.6, *p =* 0.004), high *ESR2* was associated with longer OS (HR 0.5, 95% CI 0.2–1, *p =* 0.048) and low expression of *RB1* was independently associated with shorter OS (HR 1.9, 95% CI 1.1–3.2, *p =* 0.014). (4) Conclusions: AR, ESR, and TSG expression signatures, as well as *ARV7*, *RB1,* and *ESR2* expression, have a prognostic value in mHSPC patients treated with ADT+DX.

## 1. Introduction

Prostate cancer (PC) is ranked second in cancer incidence and represents the fifth cause of cancer death in men worldwide [1]. Androgen deprivation therapy (ADT) in combination with docetaxel (DX) or anti-androgen receptor therapies (ART) are standard upfront treatments in metastatic hormone-sensitive prostate cancer (mHSPC) based on meaningful improvement in overall survival (OS) compared to ADT alone [2,3,4,5,6,7,8]. However, metastatic PC remains an incurable disease with heterogeneous clinical evolution, and treatment selection for individual patients remains a challenge. Recently, it has been shown that the transcriptional profile of primary tumors may determine a distinct clinical evolution of mHSPC patients treated with ADT alone or ADT+DX [9].

Molecular alterations in several genes such as those AR-related tumor suppressor genes (TSG) (*RB1*, *PTEN* and *TP53*), DNA-repair genes [10,11], and cell plasticity (neuroendocrine (NE), epithelial-mesenchymal transition (EMT))-related genes [12,13,14], have been associated with treatment resistance and aggressive clinical evolution of metastatic castration-resistant prostate cancer (CRPC). We hypothesized that if gene expression deregulation on those genes were present in non-castrated tumors, they could also predict clinical evolution and treatment benefit. To test this hypothesis, we designed a custom expression panel of 184 genes that may be relevant in PC biology, including genes commonly altered in CRPC, and tested it in tumor biopsies from patients with mHSPC.

We present here the results of a multicenter retrospective biomarker study in mHSPC patients treated with ADT+DX as standard clinical practice in different hospitals in Spain. The ultimate goal was the identification of gene expression signatures related to adverse outcomes that could identify patient candidates for the exploration of novel treatment strategies.

## 2. Materials and Methods

### 2.1. Design, Patients and Samples

This is a multicenter retrospective biomarker study in patients with mHSPC. Key inclusion criteria were prostate adenocarcinoma diagnosis with available formalin-fixed paraffin-embedded (FFPE) biopsy of the primary tumor or a metastatic site in the hormone-sensitive setting that was considered by the pathologist to have enough material for molecular analysis Treatment for mHSPC was ADT (i.e., luteinizing hormone-releasing hormone (LHRH) analogs) in combination with DX (75 mg/m^2^ in combination with prednisone 10 mg/day every 21 days for six cycles). Patients with primary NE tumors were excluded. Clinical variables were collected from patients’ electronic records. The volume of disease was defined according to the CHAARTED trial criteria, which considers the presence of visceral metastases or ≥4 bone lesions with ≥1 outside the spine or pelvis as high-volume disease [2].

The primary endpoint of the study was to correlate the gene expression profiles with CRPC-free survival (CRPC-FS). Secondary endpoints included overall survival (OS) and response to treatment.

### 2.2. Formalin-Fixed Paraffin-Embedded Tissue Preparation

Tissue samples were fixed in 10% neutral buffered formalin. For small biopsy samples, 6 h of fixation was required, and 12–48 h was required for surgical resection. Samples were then processed in a fluid-transfer advanced automatic tissue processor. To create paraffin blocks, a tissue embedding center (HistoStar, Thermo Scientific, Runcorn, Cheshire, UK) that contained a paraffin reservoir and dispenser, as well as warm and cold plates, was used. The first step was to pour melted paraffin until the stainless-steel mold was partially filled. The tissue samples were removed from the plastic cassettes and transferred into the bottom of the mold (the cutting surface faced down) on the warm plate. Then the tissue was oriented and pressed using a HistoPress. The labeled plastic cassette was placed on top of the mold. Finally, the blocks were cooled on the cold plate and detached from the mold. Once the paraffin-embedded blocks were made, histological sections could already be performed.

### 2.3. Gene Expression Panel Design

We configured a gene expression nCounter panel (Nanostring Technologies, Seattle, WA, USA) representing signatures described to be related to CRPC development and androgen suppression or taxane resistance [10,12,13,15,16,17,18,19,20,21,22]. The panel consisted of 184 genes, including 5 housekeeping genes (*ACTB*, *GAPDH*, *GUSB*, *HPRT1*, and *RPL13A*), and a total of 192 probe sets, with 2 site-specific probes for the isoforms III and VI of *TMPRSS2-ERG* and 8 probes for the detection of *ERG* gene expression imbalance between the 3′ and 5′ regions of mRNAs, allowing recognition of any fusion of *TMPRSS2-ERG* (Appendix A).

### 2.4. RNA Extraction

Formalin-fixed paraffin-embedded sections of PC tissues were examined with hematoxylin and eosin staining to determine the tumor area. Macrodissection was performed to avoid contamination with stroma or normal prostatic tissue. At least two 10 μm FFPE slides were used to extract total RNA by using the AllPrep DNA/RNA FFPE Kit (QIAGEN, Hilden, Germany) according to the manufacturer’s instructions. RNA was quantified by a Nanodrop Spectrophotometer ND-1000 (Thermo Scientific, Wilmington, MA, USA).

### 2.5. Gene Expression Analysis

A minimum of ~100 ng of total RNA was used to measure gene expression using the nCounter platform according to the manufacturer’s protocol (Nanostring Technologies, Seattle, WA, USA). Briefly, RNA was hybridized into 192 probe sets for 18 h at 65 °C. Samples were then processed in an automated nCounter Prep Station and imaged on a nCounter Digital Analyzer (Nanostring Technologies, Seattle, WA, USA). Raw expression counts (Appendix A) were collected, normalized, and log2 transformed using the nSolver 4.0 software. Counts normalization steps consisted of background thresholding of the mean of negative control probe counts +2 standard deviations, normalization by a factor obtained from the geometric mean of the positive control probe counts, and finally a normalization by a factor obtained from the geometric mean of the housekeeping probe counts.

*TMPRSS2-ERG* expression was assessed by the imbalance of eight *ERG* probes, four at 3′ and four at 5′. We represented the ratio between the mean of *ERG* 3′ probe counts and the mean of *ERG* 5′ counts (*ERG* 3′/5′) for each patient. These ratios were compared with previous real-time quantitative reverse-transcription PCR data of the isoform III of the *TMPRSS2-ERG* gene obtained from 77 RNA samples analyzed in a previous study [23] and the counts from a site-specific probe for the isoform III in these patients. A score threshold for the ratio *ERG* 3′/5′ of 3.4 was established to consider the presence of *TMPRSS2-ERG* alteration (Appendix A).

### 2.6. Bioinformatics and Statistical Analysis

Hierarchical cluster analysis of the expression values of the whole gene panel (excluding specific and imbalance *TMPRSS2-ERG* probes) or the signatures was performed using Cluster 3.0 [24], and results were visualized in Java TreeView [25].

Tertiles were applied to gene expression data to categorize the samples as high-, middle-, or low-expression groups. Clinical variables such as stage at diagnosis, Gleason at diagnosis, the presence of visceral metastasis, bone metastasis, the disease volume at ADT start time, and the time from ADT to docetaxel (<3 vs. ≥3 months) were evaluated as dichotomic. Lactate dehydrogenase (LDH) levels were evaluated as a continuous variable.

CRPC-FS, calculated from the date of start of ADT to the time of developing CRPC, and OS, calculated from the date of start of ADT to the time of death or last follow-up visit, were analyzed by the Kaplan–Meier method and compared by log-rank test. CRPC-FS definition, treatment-response criteria and progressive-disease definitions followed Prostate Cancer Working Group 2 criteria [26]. Univariate analysis of variables of interest was performed by Cox regression analysis; *p* < 0.1 was required for inclusion in the multivariate model. When considering all the individual genes of a signature, their expression levels were evaluated as continuous variables, and significant genes were selected if they accomplished a false discovery rate (FDR) < 0.2. Fisher’s exact test and the Wilcoxon Mann–Whitney test were used to compare the proportions of qualitative and continuous clinical variables between groups, respectively. Correlations between expression levels as continuous variables were measured by calculating Pearson’s coefficient. Significant differently expressed genes between groups were selected if they accomplished a fold change (|FC|) ≥ 1.5 and FDR < 0.05.

In order to compare the expression of a signature between groups, single-sample GSEA (ssGSEA) [27] from the GSVA R package [28] was used to calculate a gene-set-enrichment score per patient for each group, and a Wilcoxon Mann–Whitney test was then applied to test for statistical differences between groups.

Analyses were performed with R software (v.3.6.3) [29].

## 3. Results

### 3.1. Patients and Samples

A total of 133 patients were enrolled in this study: 125 of them were eligible, and 8 were excluded due to insufficient tumor sample (*N =* 4) or lack of RNA availability (*N =* 4). Table 1 summarizes the baseline clinical characteristics of the eligible patients. Of note, 92.8% (*N =* 116) of patients had de novo mHSPC disease, 20% (*N =* 25) had visceral metastasis, 78.4% (*N =* 98) were considered to have high-volume disease [2], and a Gleason score ≥8 was reported in 81.6% (*N =* 102) of patients. The number of patients who received ART (abiraterone or enzalutamide) as first-line treatment in CRPC was 77 (80.2%). We collected FFPE samples mostly from primary tumors (*N =* 117, 93.6%). The remaining biopsies were obtained from metastatic sites (*N =* 8, 6.4%).

### 3.2. Clinical Outcomes

The median follow-up time was 36.1 months (range 6.7–78.6), and 96 patients (76.8%) developed CRPC. Median CRPC-FS was 19.3 months (95% confidence interval (CI) 15.8–23.7), and median OS was 53 months (95% CI 40.4–72.6).

### 3.3. Gene Expression and Clinical Outcomes

#### 3.3.1. Global Gene Expression Analysis

An unsupervised hierarchical cluster of the whole studied panel grouped patients into three main groups, designed as A (*N =* 59, 44.8%), B (*N =* 40, 32%), and C (*N =* 29, 23.2%) (Figure 1A). Cluster B presented a shorter CRPC-FS (hazard ratio (HR) 1.9, 95% CI 1.1.3, *p =* 0.028, with respect to cluster C) and OS (HR 1.9, 95% CI 1–3.5, *p =* 0.04, with respect to cluster A) (Figure 1B). Cluster B showed a down-expression of 56 genes (vs. the other two clusters) including AR-related genes such as *KLK3*, *TMPRSS2* and *ARFL*, as well as *ESR1* and *ESR2* (Figure 1C). In this comparison, cluster B was independently associated with shorter CRPC-FS (HR 1.9, 95% CI 1.2–3, *p =* 0.007) and OS (HR 2.1, 95% CI 1.1–4, *p =* 0.021) (Figure 1D).

#### 3.3.2. Androgen Receptor Signature

Thirty-one AR-related genes were analyzed (Appendix A). Hierarchical non-supervised clustering classified patients in AR-low (*N =* 63, 50.4%) and AR-high (*N =* 62, 49.6%) categories (Figure 2A). AR-high group was associated with lower LDH levels (*p =* 0.009) (Appendix A). No differences in AR signature expression were observed between patients with high- or low-volume disease or de novo mHSPC vs. recurrent disease (Appendix A). Patients with high AR signature expression had longer CRPC-FS (HR 0.5, 95% CI 0.3–0.8, *p =* 0.002) and OS (HR 0.6, 95% CI 0.3–1; *p =* 0.041) than patients with low AR signature expression. AR signature was independently associated with CRPC-FS (HR 0.6, 95% CI 0.3–0.9; *p =* 0.015) (Figure 2B).

Considering AR signature individual genes as continuous variables in a multivariate analysis, high *ARV7* expression was independently associated with shorter CRPC-FS (HR 1.5, 95% CI 1.1–2.1, *p =* 0.008) and OS (HR 1.8, 95% CI 1.2–2.6, *p =* 0.004) (Figure 2C). When segregating *ARV7* expression levels according to tertiles, upper tertile *ARV7* expression was independently associated with shorter CRPC-FS (HR 1.7, 95% CI 1–2.7, *p =* 0.034) and OS (HR 1.9, 95% CI 1–3.4, *p =* 0.043) (Figure 2D). No correlation between *ARV7* levels and clinical variables was observed.

#### 3.3.3. Estrogen Receptor Signature

Estrogen receptor (ESR) signature was comprised of *ESR1* and *ESR2*. The hierarchical unsupervised cluster analysis of these genes grouped patients as ESR-low (*N =* 71, 56.8%) and ESR-high (*N =* 54, 43.2%) (Figure 3A). No association between ESR signature groups and clinical factors was found, and no differences in ESR signature expression were observed between patients with high or low-volume disease, or de novo mHSPC vs. recurrent disease (Appendix A). ESR-high was independently associated with longer CRPC-FS (HR 0.6, 95% CI 0.4–0.9; *p =* 0.019) and OS (HR 0.5, 95% CI 0.2–0.9, *p =* 0.024) (Figure 3B). Taking into account individual expression of ESR genes, upper tertile *ESR2* expression was independently associated with longer OS (HR 0.5, 95% CI 0.2–1, *p =* 0.048) (Figure 3C). *ESR1* expression was not related to clinical outcomes (Appendix A).

#### 3.3.4. Estrogen and Androgen Receptor Correlations

Since a negative regulation of AR signaling mediated by ESR2 has been documented [30,31], we decided to analyze the correlations between the expression of *ESR* and *AR* genes. A gene expression correlation matrix showed a significant negative correlation between *ESR2* and *ARV7* and a positive correlation between *ESR1*-*ESR2*, *ARFL-ARV7*, and *ESR1-ARFL* (Figure 3D).

Moreover, a hierarchical non-supervised cluster analysis distributed patients according to three main categories: ESR-low (*N =* 50, 40%), ESR-high+AR-low (*N =* 48, 38.4%); and ESR-high+AR-high (*N =* 27, 21.6%) (Figure 3E). ESR-high+AR-low was the group with longer CRPC-FS (HR 0.6, 95% CI 0.4–1, *p =* 0.045, with respect to the ESR-low group) and OS (HR 0.4, 95% CI 0.2–0.8, *p =* 0.009, with respect to the ESR-low group) (Figure 3F). When comparing the ESR-high+AR-low group vs. the other groups together, it was independently correlated with longer CRPC-FS (HR 0.6, 95% CI 0.4–1, *p =* 0.044) and OS (HR 0.4, 95% CI 0.2–0.9, *p =* 0.016) (Figure 3G).

Based on these observations, we decided to further explore if the relationship between AR and ESR could be associated with clinical outcomes by establishing *ESR/AR* expression ratios. Taking expression ratios as continuous variables, we found that high *ESR1*/*ARV7* and *ESR2*/*ARV7* were independently associated with longer CRPC-FS (HR 0.5, 95% CI 0.3–0.9, *p =* 0.031; HR 0.5, 95% CI 0.2–0.9, *p =* 0.029; respectively) and OS (HR 0.4, 95% CI 0.2–0.8, *p =* 0.012; HR 0.2, 95% CI 0.1–0.7, *p =* 0.008, respectively) (Figure 4A). Upper tertile *ESR1/ARV7* expression ratio independently correlated with longer OS (HR 0.4, 95% CI 0.2–0.9, *p =* 0.022), and upper tertile *ESR2/ARV7* correlated with longer CRPC-FS (HR 0.6, 95% CI 0.4–1, *p =* 0.04) and OS (HR 0.4, 95% CI 0.2–0.9, *p =* 0.02) (Figure 4B,C). Moreover, upper tertile *ESR2*/*ARFL* expression was independently associated with longer OS (HR 0.5, 95% CI 0.3–0.9, *p =* 0.021) (Figure 4D), while *ESR1/ARFL* did not correlate with clinical outcomes (Appendix A).

#### 3.3.5. Tumor Suppressor Gene (TP53, RB1 and PTEN) Signature

The expression of TSG, which associates with aggressive CRPC clinical evolution [11,32,33], was tested in this cohort of mHSPC patients. Hierarchical non-supervised cluster analysis classified patients as “TSG-low” (*N =* 67, 53.6%) or “TSG-high” (*N =* 58, 46.4%) categories (Figure 5A). No association between TSG signature groups and clinical factors was found, and no differences in TSG signature expression were observed between patients with high- or low-volume disease or de novo mHSPC vs. recurrent disease (Appendix A). Low *PTEN* levels correlated with the presence of visceral metastasis (*p =* 0.044) (Appendix A).

TSG-low expression was independently associated with shorter OS (HR 2, 95% CI 1–3.8; *p =* 0.044) (Figure 5B). Considering TSG individual genes as continuous variables in a multivariate analysis, low expression of *RB1* was independently associated with shorter OS (HR 1.9, 95% CI 1.1–3.2, *p =* 0.014) (Figure 5C). Moreover, the lower tertile expression of 2 out of the 3 TSG independently correlated with shorter CRPC-FS (HR 2.1, 95% CI 1.3–3.5, *p =* 0.003) and OS (HR 2.2, 95% CI 1.1–4.1, *p =* 0.018) (Figure 5D).

As alterations in TSG have been associated with low AR activity and NE dedifferentiation, we explored how these signatures were correlated in our series. *ARFL* was positively correlated with *RB1*, *TP53* and *ARV7* expression and negatively with *MYCN* and *AURKA*. Moreover, a negative correlation between *RB1* and *EZH2* expression was found. Additionally, a significant positive correlation between TSG and *ESR1* was observed (Figure 3D).

#### 3.3.6. Neuroendocrine and Other Signatures

The expression of forty-five NE-related genes was analyzed (Appendix A), and no correlation with clinical outcomes was observed (Appendix A). No correlation between other signatures or *TMPRSS2-ERG* expression and clinical outcomes was found (Appendix A).

#### 3.3.7. Joint Analysis of Gene Expression Signatures

Next, we assessed the significance of AR, ESR, and GST signatures together. The multivariate analysis including significant molecular signatures and clinical factors showed that high expression of AR (HR 0.5, 95% CI 0.3–0.8, *p =* 0.004) and ESR (HR 0.5, 95% CI 0.3–0.9, *p =* 0.011) signatures correlated with longer CRPC-FS, and high expression of ESR signature (HR 0.5, 95% CI 0.2–0.9, *p =* 0.033) correlated with longer OS (Figure 6).

## 4. Discussion

In this study, we show that the expression of AR, ESR and the TSG (*PTEN*, *RB1* and *TP53*) signatures are associated with the clinical evolution of patients with mHSPC treated with the combination of ADT+DX.

While AR overexpression and pathway activation has been demonstrated as one of the fundamental mechanisms of progression and resistance to therapy in CRPC patients [10,34,35,36], its role in HSPC has to be defined. In our series, we found that a high AR signature was associated with longer OS and, independently, predicted longer CRPC-FS. The molecular analysis of 160 mHSPC patients included in the phase III CHAARTED trial [9] that compared ADT vs. ADT+DX therapy showed that different PAM50 molecular subtypes have distinct treatment benefits: luminal B subtype was associated with a poorer prognosis on ADT alone but benefited significantly from ADT+DX (OS: HR 0.45, *p =* 0.007), in contrast to the basal subtype, which showed no OS benefit (HR 0.85, *p =* 0.58). These results were in contrast with a previous study where, in a subset of non-metastatic 315 patients, the luminal B subtype was the only group that benefited from postoperative response to ADT [37]. These discrepant results may be explained by the different patient populations included in both studies. As luminal expression profile is associated with high AR signaling and steroid hormone receptor processing [9], results from the CHAARTED trial may be in concordance with our data, showing better outcomes for patients with high AR-related expression when treated with the combination therapy. However, only 7 genes from the PAM50 gene set were represented in our signature, and for that reason, no definitive conclusions can be drawn about the PAM50 molecular subtypes in our cohort.

The AR splicing variant *ARV7*, which lacks the ligand-binding domain, may be constitutively activated in the absence of androgens and acts as a transcription factor repressing crucial tumor suppressor genes and promoting PC progression [38]. It has been recently shown that its detection by IHC correlates with poor prognosis and short response to ADT in mHSPC patients [39]. In our study, high *ARV7* expression was independently associated with shorter CRPC-FS and OS, supporting that *ARV7* also confers adverse prognosis in patients treated with combined therapy.

A novel and relevant result of our study is that the expression of the ESR signature is independently associated with a better outcome. When analyzing the significant signatures (AR, ESR and TSG) together, only the ESR signature was independently associated with CRPC-FS and OS. The ESR subfamily proteins are composed of two main subtypes of receptors, ESR1 and ESR2. *ESR1* may be expressed in prostate stem cells and is up-regulated during malignant transformation of the prostatic epithelium, in high-grade PIN, in metastatic lesions, and in CRPC. In contrast, *ESR2* is expressed at high levels in the luminal cells of the prostatic epithelium and may be partly lost in the high-grade PIN. *ESR2* may function in PC as a tumor suppression gene; it preferentially binds phytoestrogens and is likely to protect the prostate epithelium from malignant transformation [40]. Notably, when analyzing individual genes, *ESR2* was independently associated with a longer CRPC-FS and OS. In pre-clinical models, ESR2 down-regulates AR signaling [30,31] and up-regulates *PTEN* [30]. Moreover, it has been shown that androgen deprivation and/or long-term abiraterone therapy induces the loss of *ESR2* and *PTEN*, and the addition of ESR2 agonists together with abiraterone has been proposed as a strategy to sustain the expression of *ESR2* and offer some benefit to patients [41]. In our series, we also found an inverse correlation between *ESR2* and *ARV7* gene expression. Notably, a high *ESR2/ARV7* ratio was independently associated with a better clinical evolution. In pre-clinical models, *ESR2* stimulation reduced *ARV7* expression [42]. Overall, this may suggest that *ESR2* stimulation may be a potential strategy to revert or prevent ARV7-related resistance. Of note, *ESR1* expression positively correlated with *PTEN*, *TP53* and *RB1*, which may also explain the good prognosis of patients with a high-ESR signature. Globally, these results suggest that the transcriptional program associated with *ESR* regulates PC essential genes and support further investigation of its role as a biomarker and as a therapeutic target.

Alterations in the tumor suppression genes *PTEN*, *TP53* and *RB1* have been associated with aggressive clinical cancer evolution and resistance to conventional therapy in CRPC patients [11,32,33]. Few studies have investigated the role of TSG genomic alterations in HSPC. Gilson et al. explored the genomic landscape of mHSPC and found that the most prevalent mutations were located in *PTEN* or *TP53* [43]. Mateo et al. studied genomic aberrations in primary PC biopsies from patients who developed mCRPC [44]. They found that patients with lower expression of *RB1* had a worse prognosis, in concordance with our work where the low expression of *RB1* was independently associated with shorter OS. Another study of targeted sequencing TSG in localized and metastatic tumors reported that altered TSG increased with advanced disease, which was associated with an increase in the risk of relapse and death in mHSPC [32,45]. To our knowledge, this is the first study that investigates the prognostic value of mRNA expression of TSG in mHSPC. We found that the low expression of TSG signature was independently associated with shorter OS. Considering individual TSG genes as continuous variables, low expression of *RB1* was independently associated with shorter OSand the lower tertile expression of 2 out of the 3 TSG independently correlated with shorter CRPC-FS and OS. Our results support that the transcriptomic analysis of TSG may define the mHSPC group of patients with aggressive clinical evolution. In that sense, there is evidence that the administration of platinum-based chemotherapy may be more active than taxanes alone in aggressive mCRPC [46]. Moreover, the AKT inhibitors have shown promising results in mCRPC patients with *PTEN* alterations in combination with abiraterone [47] or DX [48]. Exploring these strategies in mHSPC with TSG alterations may be warranted.

In our study, only 6% of patients were excluded from molecular analysis due to insufficient tumor samples or lack of RNA availability. A possible explanation is that an inclusion criterion for participation in the study was to have available FFPE samples that were considered by the pathologist to have enough material for molecular analysis. Moreover, the use of the nCounter technology may also represent an advantage over other methodologies for FFPE sample molecular analysis. Our laboratory has much experience in the use of the nCounter technology in the study of transcriptional signatures in breast cancer [49] and other tumor types [50]. This technology has demonstrated high profitability for the analysis of mRNA from FFPE-tumor samples with low RNA quantity and high reproducibility [49,51,52], which has led to its clinical application in breast cancer [49]. To expand its investigation into prostate cancer is warranted.

The main limitation of this work relied on the lack of independent validation of the results. In addition, the combination of ADT+ART is another current standard treatment for patients with mHSPC and has not been explored in the present study. However, due to the potential interest that they could arise in other groups, we presented these results while we were working on independent series of mHSPC patients receiving different treatment strategies in order to validate its prognostic value and to explore its potential usefulness for treatment selection.

## 5. Conclusions

Our study suggests that *AR* and *ESR* signatures and *ESR2* gene expression correlate with good prognosis in patients receiving ADT+DX. Moreover, the lower expression of TSG (*PTEN*, *TP53* and *RB1*) signature, as well as high *ARV7* and low *RB1* gene expression, are associated with adverse clinical outcomes. The usefulness of transcriptomic analysis of such signatures as a strategy for personalized treatment selection should be further explored.

## Figures and Tables

**Figure 1 cancers-14-04757-f001:**
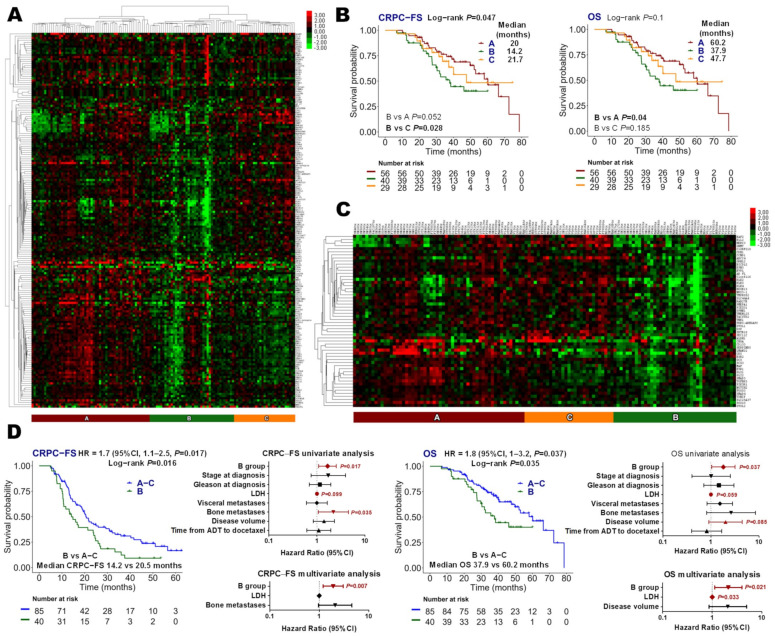
Global gene expression analysis. (**A**) Hierarchical clustering expression heatmap for expression values of the global genes panel, excluding *TMPRSS2-ERG*-specific and imbalance probes; (**B**) Kaplan–Meier curves representing CRPC-free survival (CRPC-FS) and overall survival (OS) according to groups defined from the global genes panel; (**C**) hierarchical clustering expression heatmap for expression values of differentially expressed genes in group B vs. groups A–C (|FC| ≥ 1.5 and FDR < 0.05); (**D**) Kaplan–Meier curves representing CRPC-FS and OS according to group B and groups A–C and forest plots representing the univariate and multivariate analysis. LDH: lactate dehydrogenase; HR: hazard ratio; CI, confidence interval.

**Figure 2 cancers-14-04757-f002:**
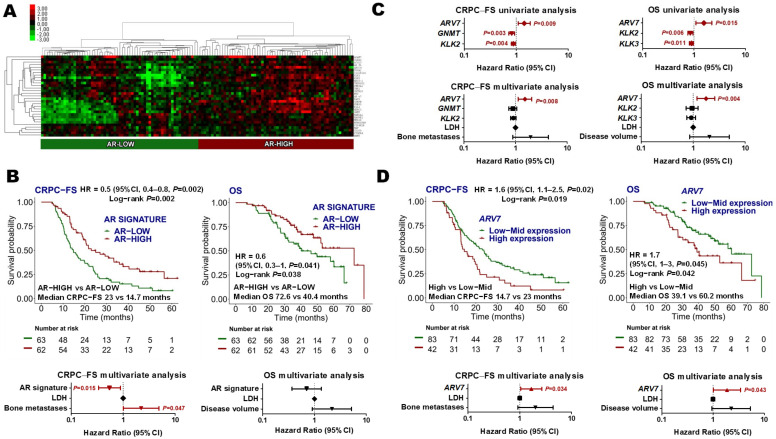
Androgen receptor (AR) signature. (**A**) Hierarchical clustering expression heatmap for expression values of the AR signature; (**B**) Kaplan–Meier curves representing CRPC-free survival (CRPC-FS) and overall survival (OS) according to AR signature and forest plots representing the multivariate analysis; (**C**) forest plots representing the univariate and multivariate analysis of the individual genes of the AR signature as continuous variables for CRPC-FS and OS. An FDR < 0.2 was applied (only genes that accomplished this condition are represented in the univariate forest plot); (**D**) Kaplan–Meier curves representing CRPC-FS and OS according to *ARV7* expression segregated into tertiles and forest plots representing the multivariate analysis. LDH: lactate dehydrogenase; HR: hazard ratio; CI, confidence interval.

**Figure 3 cancers-14-04757-f003:**
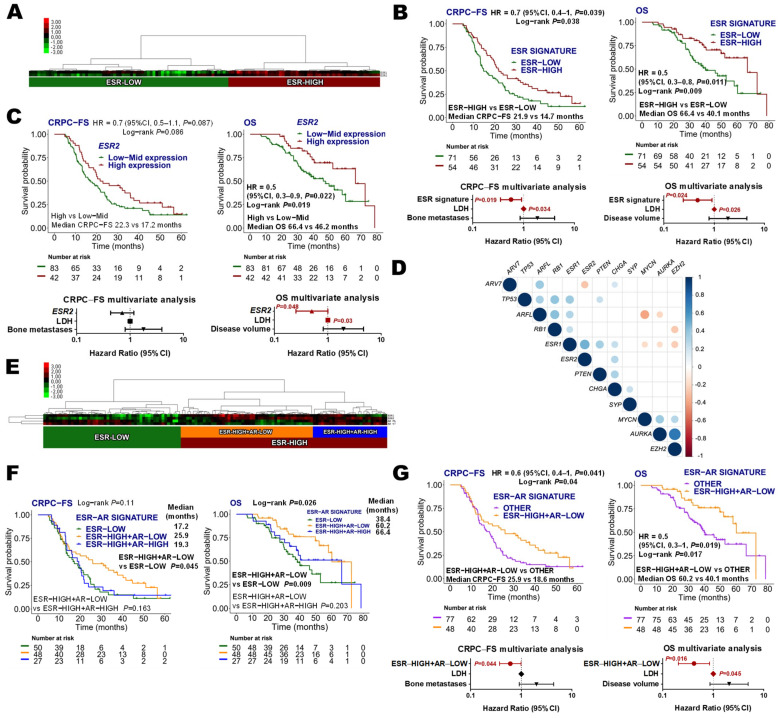
Estrogen receptor (ESR) signature. (**A**) Hierarchical clustering expression heatmap for expression values of the ESR signature; (**B**) Kaplan–Meier curves representing CRPC-free survival (CRPC-FS) and overall survival (OS) according to ESR signature and forest plots representing the multivariate analysis; (**C**) Kaplan–Meier curves representing CRPC-FS and OS according to *ESR2* expression segregated into tertiles and forest plots representing the multivariate analysis; (**D**) correlation matrix of *ESR*, *AR*, neuroendocrine (*AURKA*, *CHGA*, *SYP*, *MYCN*, and *EZH2*) and tumor suppressor (*TP53*, *RB1* and *PTEN*) genes. Correlation coefficients (r) between expression values are represented when *p* < 0.05; (**E**) hierarchical clustering expression heatmap for expression values of the *ESR* and *AR* genes; (**F**) Kaplan–Meier curves representing CRPC-FS and OS according to *ESR* and *AR* genes; (**G**) Kaplan–Meier curves representing CRPC-FS and OS according to ESR-High+AR-low and the other groups and forest plots representing the multivariate analysis. LDH: lactate dehydrogenase; HR: hazard ratio; CI, confidence interval.

**Figure 4 cancers-14-04757-f004:**
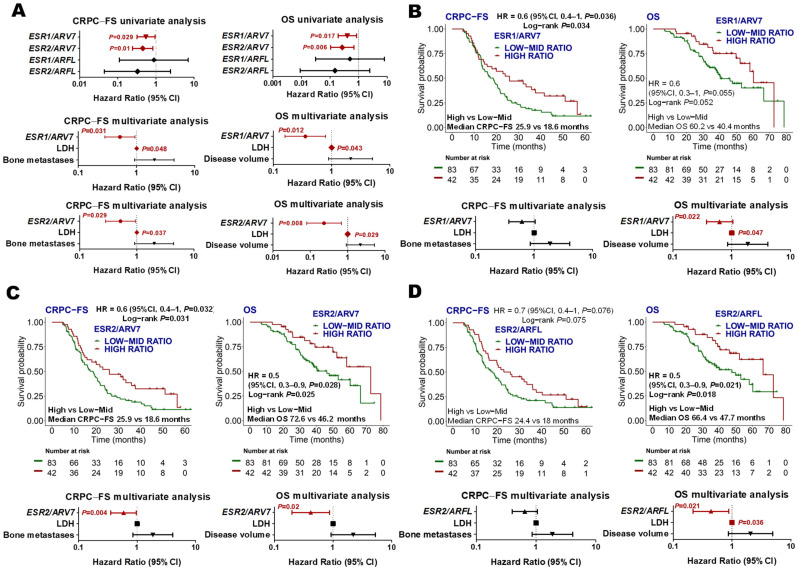
*ESR/AR* expression ratios. (**A**) Forest plots representing the univariate and multivariate analysis of the ratios between *ESR* and *AR* genes as continuous variables for CRPC-FS and OS. Those ratios with *p* < 0.1 in the univariate were included in the multivariate analysis; (B-D) Kaplan–Meier curves representing CRPC-FS and OS according to *ESR1/ARV7* (**B**), *ESR2/ARV7* (**C**), and *ESR2/ARFL* (**D**) ratios segregated into tertiles and forest plots representing the multivariate analysis. LDH: lactate dehydrogenase; HR: hazard ratio; CI, confidence interval.

**Figure 5 cancers-14-04757-f005:**
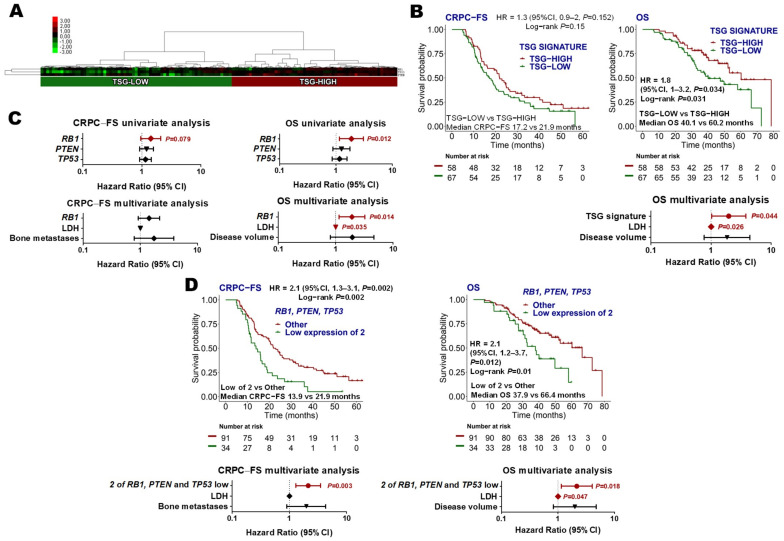
Tumor suppressor genes (TSG) signature. (**A**) Hierarchical clustering expression heatmap for expression values of the TSG signature; (**B**) Kaplan–Meier curves representing CRPC-free survival (CRPC-FS) and overall survival (OS) according to TSG signature and forest plot representing the multivariate analysis for OS; (**C**) forest plots representing the univariate and multivariate analysis of the individual genes of the TSG signature as continuous variables for CRPC-FS and OS. The reciprocal of the hazard ratio (HR) and the associated confidence interval (CI) were calculated; (**D**) Kaplan–Meier curves representing CRPC-FS and OS according to low expression of two out of the three TSG, and forest plots representing the multivariate analysis. Expression levels were segregated into tertiles to establish the cut-offs. LDH: lactate dehydrogenase.

**Figure 6 cancers-14-04757-f006:**
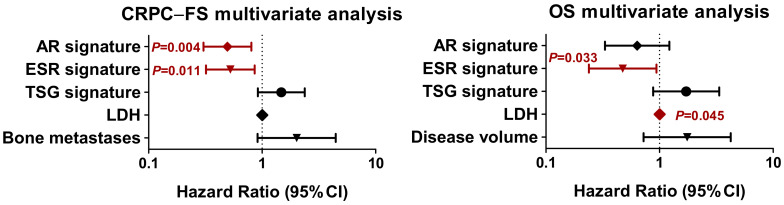
Multivariate analysis for androgen receptor (AR), estrogen receptor (ESR) and tumor suppressor gene (TSG) signatures. Forest plots represent the multivariate analysis of AR, ESR and TSG signatures for CRPC-free survival (CRPC-FS) and overall survival (OS). LDH: lactate dehydrogenase; HR: hazard ratio; CI, confidence interval.

**Table 1 cancers-14-04757-t001:** Patients’ characteristics. *N*: number of cases; ADT: androgen deprivation therapy; ECOG: Eastern Cooperative Oncology Group; PSA: prostate-specific antigen; CRPC: castration-resistant prostate cancer; NA: not available.

Patients Eligible (Enrolled), *N*	125 (133)
Age (years)	
Median (range)	66.6 (46.3–83.4)
Tumor origin, *N* (%)	
Primary	117 (93.6)
Metastatic	8 (6.4)
Stage at diagnosis, *N* (%)	
<IV	9 (7.2)
IV	116 (92.8)
Gleason sum at diagnosis, *N* (%)	
≤7	22 (17.6)
≥8	102 (81.6)
NA	1 (0.8)
Presence of bone metastases, *N* (%)	
Yes	112 (89.6)
No	13 (10.4)
Presence of visceral metastases, *N* (%)	
Yes	25 (20)
No	100 (80)
Location of visceral metastases, *N* (%)	
Lung	20 (80)
Liver	7 (28)
Pleural	1 (4)
NA	1 (4)
Disease volume, *N* (%)	
High	98 (78.4)
Low	26 (20.8)
NA	1 (0.8)
ECOG performance status score, *N* (%)	
0	54 (43.2)
1 or 2	69 (55.2)
NA	2 (1.6)
Baseline PSA (ng/mL) at diagnosis	
Median (range)	83.2 (1.8–7448)
Baseline lactate dehydrogenase (U/L)	
Median (range)	316 (116–1023)
Time from ADT to docetaxel treatment, *N* (%)	
<3 months	106 (84.8)
≥3 months	19 (15.2)
First line treatment in CRPC, *N* (%)	
Abiraterone or enzalutamide	77 (80.2)
Taxanes	5 (5.2)
Other treatments	5 (5.2)
No treatment	4 (4.2)
NA	5 (5.2)

## Data Availability

The raw counts from NanoString nCounter gene expression data generated in this study are available in Appendix A.

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
