# Peer review of "Transcriptional Profile Associated with Clinical Outcomes in Metastatic Hormone-Sensitive Prostate Cancer Treated with Androgen Deprivation and Docetaxel"

_cancers, 2022, doi:10.3390/cancers14194757_

Round 1
Reviewer 1 Report
It is important to recognize the great work done by the authors. As you comment, an independent validation of your results would be important. The study design is adequate and the results are interesting to open new lines of research.
Reviewer 2 Report
The authors evaluated the transcriptional profile associated with clinical outcomes in metastatic mHSPC treated by ADT+DX. The results are interesting and informative. However, the manuscript needs several major revisions.
1. Page 4, line 156. The quality of RNA is very important to evaluate the gene expression profiles. The RNA was extracted from FFPE local prostate biopsy specimens in most cases. It might be difficult to keep RNA with high quality in old FFPE samples. In this study, the samples were retrospectively collected in multiple institutes. However, only 8 out of 133 patients were excluded due to insufficient tumor samples of lack of RNA availability. The methods to extract RNA was described in page 3, line 102. The methods to prepare the FFPE samples should be described in detail.
2. Page 4, line 165. The CRPC-FS and OS were evaluated in the subgroups based on the gene expression profiles. All the patients received ADT+DX therapy as the first line treatment. The number of patients receiving second line androgen receptor axis targeting therapy (ARAT) such as abiraterone or enzalutamide should be described.
3. Page 12, line 399. The standard first line treatment for metastatic prostate cancer patients changed form ADT+DX into ADT+ARAT. The authors should describe them as a limitation of this study.
Reviewer 3 Report
In this manuscript, Jiménez et al. conducted bioinformatic analysis on the gene signatures of metastatic hormone-sensitive prostate cancer (mHSPC) patients treated with androgen deprivation therapy (ADT) and docetaxel (DX). Based on patient treatment (ADT+DX) data in Spanish hospitals, the authors explored the correlation between commonly studied prostate cancer genes like ESR1/2, AR-FL, ARV7 and CRPC-FS/OS, as well as compiled a comprehensive list of 184 gene signature that would predict treatment outcome for prostate cancer treatment. The finding of this paper should thus be of interest to prostate cancer clinicians and basic science researchers.
Suggestions and comments:
11. Although ARv7 is one of the major AR variants in prostate cancer, the authors should also explore the relationship between other AR variants and the factors considered in this study.
22. Some abbreviations in the abstract (OS, CI, HR, CRPC-FS) are not defined (in the abstract), although they are (defined) in the main text, as such the abstract can be difficult to read for someone who is not a prostate cancer and/or a bioinformatics expert.
33. Label font size of most of the figures is very small which is a challenging test for a reader’s eyesight but certainly not ideal for a high-quality publication. Although this is a relatively minor issue, I will not recommend publication unless the font is changed to a readable size (for all figures).
Round 2
Reviewer 2 Report
The manuscript was precisely corrected.